# Optimization of Revision Hip Arthroplasty Workflow by Means of Detailed Pre-Surgical Planning Using Computed Tomography Data, Open-Source Software and Three-Dimensional-Printed Models

**DOI:** 10.3390/diagnostics13152516

**Published:** 2023-07-28

**Authors:** Krzysztof Andrzejewski, Marcin Domżalski, Piotr Komorowski, Jan Poszepczyński, Bożena Rokita, Marcin Elgalal

**Affiliations:** 1Department of Orthopaedics and Trauma, Veteran’s Memorial Hospital, Medical University of Lodz, Zeromskiego 113, 90-549 Lodz, Poland; kj.andrzejewski@outlook.com (K.A.); marcin.domzalski@umed.lodz.pl (M.D.); janek24061982@tlen.pl (J.P.); 2Division of Biophysics, Institute of Materials Science, Lodz University of Technology, Stefanowskiego 1/15, 90-924 Lodz, Poland; piotr.jerzy.komorowski@gmail.com; 3Institute of Applied Radiation Chemistry, Faculty of Chemistry, Lodz University of Technology, Wroblewskiego 15, 93-590 Lodz, Poland; bozena.rokita@p.lodz.pl; 4Second Department of Radiology and Diagnostic Imaging, Medical University of Lodz, Pomorska 251, 92-213 Lodz, Poland

**Keywords:** anteversion angle, center of rotation, inclination angle, three-dimensional models, revision hip arthroplasty, virtual planning

## Abstract

Background. In revision hip arthroplasty (RHA), establishing the center of rotation (COR) can be technically challenging due to the acetabular bone destruction that is usually present, particularly in severe cases such as Paprosky type II and III defects. The aim of this study was to demonstrate the use of open-source medical image reconstruction software and low-cost 3D anatomical models in pre-surgical planning of RHA. Methods. A total of 10 patients, underwent RHA and were included in the study. Computed tomography (CT) scans were performed for all cases, before surgery and approximately 1 week after the procedure. The reconstruction of CT data, 3D virtual planning of the COR and positioning of acetabular cups, including their inclination and anteversion angles, was carried out using the free open source software platform 3D Slicer. In addition, anatomical models of the pelvis were built on a desktop 3D printer from polylactic acid (PLA). Preoperative and postoperative reconstructed imaging data were compared for each patient, and the position of the acetabular cups as well as the COR were evaluated for each case. Results. Analysis of the pre- and post-op center of rotation position data indicated statistically insignificant differences for the location of the COR on the *X*-axis (1.5 mm, t = 0.5741, *p* = 0.5868) with a fairly strong correlation of the results (r = −0.672, *p* = 0.0982), whilst for the location of the COR in the Y and Z-axes, there was statistical dependence (Y axis, 4.7 mm, t = 3.168 and *p* = 0.0194; Z axis, 1.9 mm, t = 1.887 and *p* = 0.1081). A strong correlation for both axes was also observed (Y and Z) (*Y*-axis, r = 0.9438 and *p* = 0.0014; *Z*-axis, r = 0.8829 and *p* = 0.0084). Analysis of inclination angle values showed a statistically insignificant difference between mean values (3.9 degrees, t = 1.111, *p* = 0.3092) and a moderate correlation was found between mean values (r = −0.4042, *p* = 0.3685). Analysis of the anteversion angle showed a statistically insignificant difference between mean values (1.9 degrees, t = 0.8671, *p* = 0.4192), while a moderate correlation between mean values was found (r = −0.4782, *p* = 0.2777). Conclusions. Three-dimensional reconstruction software, together with low-cost anatomical models, are very effective tools for pre-surgical planning, which have great potential use in orthopedic surgery, particularly RHA. In up and in- and up and out-type defects, it is essential to establish a new COR and to identify three support points within the revision acetabulum in order to correctly position acetabular cups.

## 1. Introduction

In recent years, there has been a significant increase in the number of primary total hip arthroplasty (THA) procedures performed worldwide; not only in the elderly, but increasingly in patients under the age of 65, with the greatest increase in patients between 45 and 54 years of age [1,2,3]. Consequently, the number of revision hip arthroplasty (RHA) procedures, due to various complications [3], has also increased. In order to improve the long-term survival of primary THA and thus reduce the need for revision procedures, several approaches have been used, e.g., enhanced biocompatibility and osteointegration of biomaterials used in the production of implants and by reduced contact pressure and wear of implant surfaces [4,5,6,7,8,9].

However, clinically, first and foremost, attempts are made to select the correct implant size (i.e., acetabular cup, femoral head and stem) and accurately plan their positions in primary THA [1,7,8,9]. Unfortunately, this is challenging during surgery and often not optimally achieved, especially by less experienced surgeons. This results in complications such as dislocation, periprosthetic fractures, aseptic loosening, etc., which ultimately leads to THA revision operations. Consequently, in recent times, there has been a greatly increased focus on accurate reconstruction of the center of rotation, particularly in difficult primary hip arthroplasties as well as in RHA [1,2,3,10,11,12,13]. It is a well-known fact that restoration of the center of rotation in the primary and revision hip arthroplasty is beneficial in balancing the stress distribution of the hip joint, which in turn reduces the wear of polyethylene inserts and prolongs the life of the endoprostheses.

The classic methods of determining the COR [10] are mainly based on 2D X-ray imaging and are subject to errors or have certain limitations (Figure 1) [14,15,16,17,18,19].

A good way to solve this problem is the use of volumetric medical imaging, reconstruction software, and virtual planning together with 3D printing and anatomical models [1]. By combing these diverse techniques into one process, it is possible to take advantage of the recent developments in these technologies and use them in the field of medicine, in particular, surgery. Detailed medical imaging in the form of Computed Tomography or Magnetic Resonance Imaging (MRI) together with appropriate software, enables surgeons to perform virtual planning of complicated operations. This process can be further enhanced through the use of patient specific anatomical models created using additive manufacturing methods, which in recent times have become much more accessible with the rapid development of budget desktop 3D printers. With such models, surgeons are able to physical assess the size and shape of the affected pelvis, which in turn makes it easier to classify the type of defect present, plan the use of metal augments or allografts (Figure 2) and any necessary bone screws. Furthermore, senior doctors can use these anatomical models as a teaching tool [20] when training junior colleagues to perform complex procedures, which is an essential part of the learning process for future surgeons. Finally, 3D anatomical models have been shown to be effective devices for patient education [21] due to the fact that they can visualize their specific conditions. Despite these numerous advantages, to date, these techniques have not been commonly used in medicine, especially in the everyday clinical practice of orthopedics. Despite the numerous developments of software used in the analysis and reconstruction of different medical images (i.e., CT, MRI) as well as advances in additive technologies, to date, these techniques are not commonly used in medicine, especially in the everyday clinical practice of orthopedics. This is all the more puzzling considering the possibilities offered by current surgical planning software and 3D printers in the planning and evaluation of surgical procedures. One of the reasons for this fact may be the cost of purchasing specialized software and high-end 3D printers. Another reason may be the difficulty of learning and using such software, as well as the complexity of some types of additive technologies. The solution to this problem may be the use of freely available, open-source software and relatively simple Fused Deposition Modeling (FDM)/Fused Filament Fabrication (FFF) 3D printers. This could provide a low-cost, relatively easy-to-learn solution to performing presurgical planning for complex procedures such as revision hip arthroplasty.

The aim of this study was to demonstrate a practical and straightforward method to plan revision hip arthroplasty procedures, using open-source software in conjunction low-cost desktop FFF 3D printers that can produce significantly cheaper anatomical models [22], compared to other additive manufacturing technologies such as SLA, PolyJet or FDM.

## 2. Materials and Methods

### 2.1. Clinical Information

Between January 2021 and July 2022, a total of 10 patients (4 male and 6 female), aged 50–65 years with an average age of 60 years, who underwent revision HA were included in the study. The study population, although small, was homogeneous, with the exception of one patient (a single case of up and out type defect). All patients had undergone primary unilateral THA with cemented acetabular cups that had become loosened, Paprosky type II and IIIA (i.e., vertical upward migration > 3 cm and intact Kohler’s line) acetabular defects. For each patient a posterior approach was carried out using the Kocher-Langenbeck—lateral position. In order to prevent injury to the sciatic nerve, the ipsilateral knee was kept in a flexed position during the procedure. Primary endoprosthesis stem removal was required in any of the cases. Standard antibiotic prophylaxis with Cefazolin was administered. Intraoperative X-ray fluoroscopy was not used, however radiographic imaging was performed immediately after skin closure. All patients were operated on by the same surgical team and orthopedic department. For all patients, the same implant system was used, i.e., Trabecular Metal Acetabular Revision System (Zimmer Biomet, Warsaw, IN, USA). In case of insufficient contact of the revision acetabulum with the pelvic bone, an allograft was used. This study was approved by the Ethics Committee on Human Research of the Medical University of Lodz, No. RNN/126/22/KE. 

### 2.2. Determining Center of Rotation, Inclination and Anteversion Angles

Computed Tomography (CT) scans were performed for all subjects, preoperatively and approximately 1 week after surgery, using one of the following scanners: Siemens Sensation Cardiac 64 (Siemens Healthcare, Erlangen, Germany) or GE Medical Systems Optima CT540 (GE Healthcare, Chicago, IL, USA), according to a predetermined CT study protocol (Table 1).

CT data were saved as DICOM files (Figure 2A). Segmentation of bone structures (Figure 2B) was carried out using the free open-source software platform 3D Slicer 5.0.2 [24]. The center of rotation, inclination and anteversion angles were established using the following workflow (Figure 3):

Step 1: Three planes were created, the first (Sagittal plane) passing through the center of the pubic symphysis and the center of the sacrum (Figure 2C). Next, a second plane was created between the tips of the ischial tuberosities (Axial plane), and finally a third plane (Coronal plane) was formed between the tips of the ischial spines on both sides.

Step 2: A sphere was drawn with dimensions corresponding to the intact side femur head and its center aligned with the center of the femur head. The distance, at right angles, of the center of this sphere relative to all three planes created in Step 1 was then measured.

Step 3: A second sphere with identical dimensions was drawn on the contralateral side, i.e., the revision side. This sphere was positioned with its center at identical distances to the three planes created in Step 1 to those of the sphere on the intact side that was aligned with the femur head. Thus, a new center of rotation was established, with XYZ coordinates based on the COR of the contralateral uninjured femur head (Figure 2C).

Step 4: the inclination and anteversion angles were determined. The Axial plane was used to determine the inclination angle with values ranging from 30 up to 45 deg. (Figure 2D). The Coronal plane was used to determine the angle of anteversion with values ranging from 10 up to 20 deg. (Figure 2D).

### 2.3. 3D Virtual Models of Acetabular Cups

A set of acetabular cup 3D models, which were used for presurgical planning, were created using CAD/CAM software (FreeCAD 0.20). The model sizes and design were based on the general shape and dimensions of the Trabecular Metal Acetabular Revision System (Zimmer Biomet, USA). The set of models consisted of acetabular cups with a diameter range of 50–64 mm, with an incremental increase in diameter of 2 mm. The final 3D models were exported and saved as *.stl* files.

### 2.4. Virtual Presurgical Planning and Post-Op Assessment of the Position of the Acetabular Cup

The virtual planning of the revision surgery was carried out in 3D Slicer 5.0.2. After determining the center of rotation together with the inclination and anteversion angles (Figure 4A), virtual planning of the RHA was performed. This process took into account the size and position of the acetabular cup, as well as the position of all the fixing screws. At first, the damaged acetabular cup was removed from the 3D model of the pelvis. The shape and dimensions of the resulting bone tissue defect were then estimated. Based on this analysis, an optimal revision acetabular cup size was determined. Then, a 3D model of the selected acetabular cup size was imported into 3D Slicer. This model was then virtually positioned within the planned implantation site, taking into consideration the center of rotation, inclination and anteversion angles (Figure 4B–D,F). After surgery and the acquisition of postoperative CT data, the position of the real acetabular implant was determined and compared to the location of the virtual 3D presurgical model (Figure 4D).

### 2.5. 3D Printing of Pelvic Models

CT data were segmented and the resulting 3D models were exported as *.stl* files and later imported into the 3D printing software CURA 5.0.0 (Ultimaker B.V, Utrecht, The Netherlands), which was used to configure the printing process. Next, physical models were made using an Ultimaker U2+ 3D printer with a 0.4 mm nozzle, layer resolution of 200-20 microns and XYZ resolution of 6.9 × 6.9 × 2.5 microns (Ultimaker B.V, Utrecht, The Netherlands) from polylactic acid (PLA) (Ultimaker B.V, Utrecht, The Netherlands), using predetermined printing parameters (Table 2) (Figure 4G). Following the printing process, support material was removed, then models were cleaned and dried.

### 2.6. Accuracy Analysis of 3D Models

Analysis of the geometric dimensions of the 3D-printed models in relation to the anatomical features of the patient’s revision acetabulum was performed by measuring the length of the pelvic bone defect in the axial plane using an electronic caliper (0–150 mm × 0.01 mm) (Techsam-Quatros, Lublin, Poland) for the 3D models, whilst a rule prepared from K-wire and measured with a caliper during the revision operation. In addition, the virtual model was measured using the Markups module in the 3D Slicer 5.0.2 software (Figure 5).

### 2.7. Statistical Analysis

OriginPro software 2022 (OriginLab Corporation, Northampton, MA, USA) was used for statistical analysis. The results are presented as mean ± standard deviation (SD) and Kolmogorov–Smirnov test was used to test for normality. Differences between the pre-surgical plan and post-op results were evaluated by 2-sided paired *t*-test. Correlations between the variables were determined using the Pearson correlation coefficient. Differences between mean values were considered as statistically relevant with a significance level of: * *p* < 0.05.

## 3. Results

### 3.1. Revision Acetabular Cup Sizes, Number and Size of Fixing Screws

The average size of the actual revision acetabular cups used during surgery was 54 ± 6 mm and in each case matched the size of the virtual, planned cup sizes. Similar results were achieved regarding the number and size of fixing screws—average number was 2 and average screw length was 35 mm.

### 3.2. Center of Rotation Position (Appendix A) and Accuracy Analysis of 3D Models (Table 3)

For each patient, a set of measurements was acquired in all three axes. The preoperative distance represents the position of the original (i.e., primary THA) acetabular cup relative to the three planes. The 3D planning distance represents the planned position of the revision acetabular cup based on the parameters of the healthy, contralateral hip joint. Lastly, the post-operative distance represents the actual, final position of the acetabular cup following surgery, evaluated on the basis of follow-up CT data.

For the pre-operative locations of the primary acetabular cups, the mean values for the COR were 86.3 ± 8.7 mm in the X axis (Coronal plane), 71.7 ± 9.6 mm in the Y axis (Coronal plane) and 52.2 ± 4.8 mm in the Z axis (Axial plane). Mean values for the COR, resulting from the 3D planning process, were 86.0 ± 7.8 mm in the X axis (Coronal plane), 71.3 ± 11.8 mm in the Y axis (Coronal plane) and 51.8 ± 2.3 mm in the Z axis (Axial plane). After surgery, mean values for the final COR position were 87.5 ± 8.6 mm in the X axis (Coronal plane), 66.6 ± 10.5 mm in the Y axis (Coronal plane) and 53.7 ± 4.5 mm in the Z axis (Coronal plane) (Appendix A).

The 3D planning values of the COR location compared to those obtained after surgery indicated statistically insignificant differences in the X axis (t = 0.5741, *p* = 0.5868) with a fairly strong correlation of results (*r* = −0.672, *p* = 0.0982). Whilst, for the Y axis and the Z axis there was a statistical relationship (Y axis, *t* = 3.168 and *p* = 0.0194; Z axis, *t* = 1.887 and *p* = 0.1081) and a strong correlation of values was also observed for both axes (Y and Z) (*Y*-axis, *r* = 0.9438 and *p* = 0.0014; *Z*-axis, *r* = 0.8829 and *p* = 0.0084) (Appendix A).

For each patient, three sets of measurements for the inclination and anteversion angles were also obtained (Appendix A). The preoperative mean inclination angle was 55.7 ± 15.4 deg; for 3D planning, the average inclination angle was 47.7 ± 5.6 deg., whilst the postoperative mean inclination angle was 51.6 ± 5.3 deg. The pre-surgical 3D planning values of the inclination angle values compared to those obtained from postoperative CT data showed a statistically insignificant difference between the mean values (*t* = 1.111, *p* = 0.3092). A moderate correlation was found between the values (*r* = −0.4042, *p* = 0.3685) (Appendix A).

For the anteversion angle, the preoperative mean value was 21.1 ± 9.7 deg, 3D planning 17.3 ± 4.1 deg and postoperative 15.4 ± 5.2 deg. The 3D planning values of the anteversion angle values compared to those obtained from the postoperative CT data showed a statistically insignificant difference between the mean values (*t* = 0.8671, *p* = 0.4192); however, a moderate correlation was found between the values (*r* = −0.4782, *p* = 0.2777) (Appendix A).

Basic analysis of the geometric parameters of the 3D models in relation to the 3D virtual models showed an average difference of −0.12 mm ± 0.24, which corresponds to a difference of 99.76% ± 0.47, while the verification of the 3D models in relation to the damaged pelvic bone showed an average difference of 0.27 mm ± 0.55, which corresponds to a difference of 100.55% ± 1.09, and in the case of the analysis of geometrical parameters of 3D virtual models in relation to the damaged pelvic bone, it showed an average difference of 0.15 mm ± 0.37, which corresponds to a difference of 100.31% ± 0.70 (Table 3).

**Table 3 diagnostics-13-02516-t003:** Accuracy analysis of 3D models (mean ± SD, *n* = 10).

Information about Types of Defect (Paprosky Classification)	3D Virtual Models Measurement/ 3D Model Measurement Results mm (% Difference)	3D Model Measurement/ Pelvis of Patient Measurement Results mm (% Difference)	3D Virtual Model Measurement/ Pelvis of Patient Measurement Results mm (% Difference)
Patient 1 (Paprosky IIB)	−0.15/99.66	0.25/100.56	0.10/100.23
Patient 2 (Paprosky IIC)	−0.04/99.92	0.10/100.19	0.06/100.12
Patient 3 (Paprosky IIB)	−0.11/99.80	0.16/100.30	0.05/100.09
Patient 4 (Paprosky IIA)	−0.70/98.59	1.50/103.12	0.80/101.67
Patient 5 (Paprosky IIB)	−0.21/99.63	0.81/101.45	0.60/101.07
Patient 6 (Paprosky IIA)	0.05/100.09	0.51/100.93	0.56/101.02
Patient 7 (Paprosky IIC)	0.21/100.36	0.40/99.32	−0.19/99.67
Patient 8 (Paprosky IIA)	−0.05/99.91	0.14/99.76	−0.19/99.67
Patient 9 (Paprosky IIB)	−0.15/99.74	0.04/99.93	−0.19/99.68
Patient 10 (Paprosky IIIA)	−0.03/99.94	0.04/99.91	−0.07/99.85
mean ± SD	−0.12 ± 0.24/99.76 ± 0.47	0.27 ± 0.55/100.55 ± 1.09	0.15 ± 0.37/100.31 ± 0.70

## 4. Discussion

Hip revision surgeries are currently one of the most challenging procedures in orthopedic surgery. According to the data, the number of scheduled hip revisions will increase by 137% in the United States, England and Wales by 2030 [25]. In comparison to primary hip arthroplasty, this type of procedure carries a much higher risk of early or late complications: up to 40% [26]. The most significant include infection, dislocation of the endoprosthesis or loosening of its elements. Currently, 2D X-ray and/or 3D CT imaging of the hip joints are used in the preoperative planning process. On the basis of these diagnostic images and using the Paprosky classification system, the type and extent of bone tissue loss can be established. Consequently, an optimal acetabular cup and, if necessary, femoral stem can be chosen. It has been proven that accurate reconstruction of the center of rotation is a key element of this surgical procedure and guarantees the best early and late post-operative outcomes [1,18,19].

Correct restoration of the COR, that is, as close to the anatomical location as possible, is crucial to achieve optimal hip biomechanics and stabilization of hip revision implants. For the damaged acetabulum, any methods that help determine the location of the acetabular center, accurately implant acetabular cups, and restore the hip rotation center have always been of great interest to orthopedic surgeons [27]. Previous studies have shown numerous techniques to locate the COR of the hip joint, including the template method, the concentric circle method of the healthy side, the Ranawat method, the Pierchon method and the Pagnano method. However, all of the above approaches are based on two-dimensional X-ray imaging of the pelvis. In addition, we know from clinical practice that removal of osteointegrated acetabular components usually results in additional bone tissue defects that are not present in preoperative imaging data. Therefore, it is not possible to precisely establish the center of rotation solely on the basis of such imaging.

Moreover, at present, there are no relatively simple, intuitive and reproducible methods for surgeons to accurately determine the hip joint center of rotation during surgery. Li J. et al. [27] observed that the accuracy of acetabular component position, in particular the degree of anteversion, could be improved by using the vertex of the Harris fossa as a reference point. However, this method did not help to establish the center of the acetabulum. Idrissi ME et al. [28] believed that the transverse acetabular ligament is a reliable anatomical marker because it is obvious, constant and not affected by factors such as change in position of the pelvis or acetabular dysplasia. Arckhold et al. [29] identified the transverse acetabular ligament in 1000 consecutive primary total hip arthroplasties and determined the position of the acetabular prosthesis based on its position with satisfactory outcomes. However, Epstein et al. [30] questioned the above method when they observed the acetabulum in 63 hip joint replacement procedures (64 hips) and determined that in 53% of such cases the transverse acetabular ligament could not be located. Additionally, Zhou JS et al. [31] determined that the transverse acetabular ligament no longer existed in most revision cases. However, their study also showed that in cases of hip revision cases, pathological changes within the inferior part of the acetabulum were relatively minor compared to the superior part, whilst remnants of the Harris fossa as well as the acetabular notches were found in these cases. Zhang H et al. [32] reported that the acetabular center was located on average 28 mm above the vertical bisection of the anterior and posterior acetabular notches line (range 25~31 mm, depending on the size of the acetabulum), and the acetabular center was located cephalad relative to the Harris fossa, close to the semilunar cartilage.

Based on the observations of other authors, it can be seen that when the acetabulum loses its normal geometry due to considerable bone defects, establishing the location of the COR can be very challenging. Although several methods have been described, as shown above, to date this has largely depended on the clinical experience of the operating surgeon. Preoperative planning, including the use of 3D-printed pelvic models, to vary the size and position of acetabular augments has been shown to be beneficial. In different cases of complex acetabular bone loss, CT with 3D reconstruction can help the surgeon determine the size and shape of any tissue defects present, as well as help determine the size and shape of any necessary augments [33]. Furthermore, the use of 3D preoperative planning can improve cup positioning in total hip arthroplasty by increasing the accuracy of anteversion restoration and reducing the percentage of outliers.

For these reasons, the authors sought to use open-source software [24] and low-cost 3D printers to provide surgical teams with additional tools to plan revision THA procedures. In our study, we analyzed the location of the COR in a three-dimensional, multiplanar environment, which was possible due to the use of volumetric CT imaging data. Detailed segmentation of bone structures and the introduction of predefined planes into the spatial configuration (Figure 2 and Figure 4) made it possible to determine the center of rotation of the operated side, based on COR position of the contralateral unaffected hip joint, using numerical data in three axes i.e., X, Y and Z [10]. Such detailed 3D models also allowed surgeons to identify three support points for the acetabular cup, within the revision acetabulum. For each case, after surgery and follow-up CT imaging, we focused on comparing the position of the COR in the presurgical plan, with the final post-op location based on follow-up imaging data. The difference in position of the COR in the X axis was 1.5 mm, which was not statistically significant and demonstrated a fairly strong correlation of results (*r* = 0.6720, *p* = 0.0982). The differences in the Y and Z axes were 4.7 mm (*r* = 0.9438, *p* = 0.0014) and 1.9 mm (*r* = 0.8829, *p* = 0.0084), respectively, which was statistically significant and demonstrated a strong correlation of results. These results indicate that the surgical team was better prepared for the procedure and pre-surgical planning resulted in accurate positioning of the revision acetabular cup.

Other important acetabular positioning parameters are the angles of anteversion and inclination. In our study, we observed statistically insignificant differences between mean values for both angles (for the inclination angle the difference was 3.9 degrees and for the anteversion angle 1.9 degrees), with moderate correlation of results (r = 0.4042, *p* = 0.3685 for the inclination angle and r = 0.4782, *p* = 0.2777 for the anteversion angle). Such results were due to the various degrees of bone damage and acetabulum displacement in the study group. Other authors attributed these differences to the position of the patient during the procedure, different methods of assessing the inclination angle [34], and the surgeon’s natural tendency to implant the acetabulum with a higher anteversion angle due to inaccurate visual interpretation of the acetabulum during surgery [35]. Finally, the accuracy of 3D-printed bone models is important for planning and revision surgery [36,37,38] Since there are many studies describing the process of verifying the accuracy of 3D-printed bone models, we decided to limit ourselves to a basic analysis of the geometric parameters of physical 3D models in relation to 3D virtual models and anatomical pelvic bone. The analysis showed slight differences in the values of the measured segment and thus showed satisfactory accuracy of 3D prints, which is consistent with the published data of other authors [36,37,38].

The most important aspects of our paper was to show the possibility of using easily accessible open-source software for planning complex procedures in conjunction with 3D printed models of anatomical structures, made with inexpensive desktop 3D printers. Such a low-budget approach could provide an excellent set of tools for teams of orthopedic surgeons, especially in countries with limited resources or those working in conflict zones, where access to advanced systems is very limited whilst the number of patients requiring surgical treatment is very high. The methodology presented using 3D Slicer software together with 3D models could also be used in planning treatment of other conditions, e.g., in patients with high-riding dislocation developmental dysplasia of the hip [35,39] and even in cases where custom-made implants are used, as software for segmentation of bone structures [40]. Furthermore, the use of these techniques is not limited to pelvic surgery, but could be applied to spinal surgery for the purpose of planning pedicle insertion and osteotomy procedures in cases of spinal deformity [41], craniofacial surgery to prepare custom-made titanium mandibular protheses [42], as well as cancer treatment and the use of MRI-based three-dimensional models, e.g., planning of radical prostatectomy [43].

Unfortunately, our study has two main limitations, the first and most important being the small number of cases analyzed. As a result of the small study population, it was difficult to clearly determine a homogeneous group of patients and perform a more thorough statistical analysis. The second limitation is the quality of the 3D-printed models used in the study and, above all, accurate segmentation of pre- and post-op CT data. Detailed 3D models of the revision site that are a result of meticulous segmentation of bone structures have a significant impact on a surgeon’s ability to assess the degree of tissue deformation and/or destruction of the acetabulum, especially in patients with a primary cemented acetabulum. During the process of learning to segment and 3D-print such models, we observed a tendency, in particular among engineers, to “improve” 3D models using segmentation software smoothing algorithms. This resulted in less detailed models in which certain bone defects were difficult to discern and thus increased the risk of overlooking or underestimating the degree of bone tissue loss. With time and increased number of patient cases, and after frequent consultations with other members of the team consisting of an orthopedic surgeon, radiologist and biomedical engineer, this tendency was eliminated. An additional method that could be possibly be used to assess 3D models of the operated acetabulum and aid in establishing the COR, is the application of an alignment technique and iterative closest point algorithms. Such solutions have been shown to be effective used in trauma patients and may be helpful in pre-surgical planning of RHA [44] Another method is the use of neural network (NN) and artificial intelligence (AI) in the process of bone tissue segmentation [45]. Wu et al. [46] showed that the use of new neural networks (e.g., CNN-CMG Net) increases the accuracy of bone tissue segmentation while reducing the time required to create 3D models of such anatomical structures, which clearly indicates a potential research direction to optimize the presurgical planning process. However, it should be emphasized that these promising results focused on healthy bone tissue. The authors clearly indicated that their work did not concern segmentation of bone structures in which metal implants were present. Nevertheless, it seems that the current development of information technology based on AI is very promising, and it is only a matter of time when surgical planning for THA and RHA will be based on neural networks and possibly simplified tissue segmentation methods [47,48,49].

## 5. Conclusions

Segmentation and 3D reconstruction software, together with low-cost FDM 3D-printed anatomical models, are very effective tools that have great potential use in orthopedic surgery and presurgical planning. However, they should always be treated as additional tools in preparation for THA procedures, and surgeons must still carry out a thorough assessment of their patients, including preoperative X-ray and CT imaging. Ultimately, surgeons will make decisions based on their clinical and surgical experience.

In up and in- and up and out-type defects, it is essential to identify the three support points of the revision acetabulum. Identification of acetabular roof, ischium, pubic bone and acetabular wall defects is possible thanks to 3D printing and CT image segmentation. Determining the postoperative center of rotation based on follow-up CT data is essential for assessing the success of the operation.

In the future, we plan to carry out further research and a series of studies on the subject of revision hip arthroplasty that will also include long-term follow-up patient data. This technical note is only the first stage of this research that focuses on using a low cost solution to optimize the planning process of revision HA.

## Figures and Tables

**Figure 1 diagnostics-13-02516-f001:**
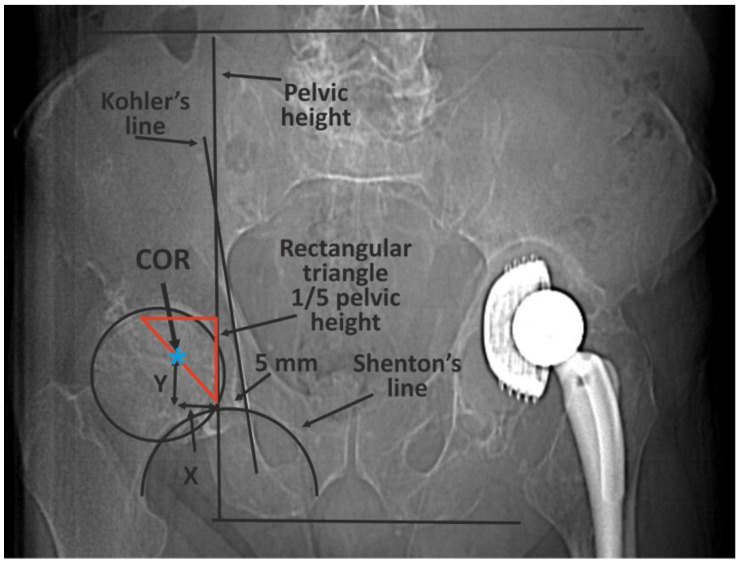
An example of the classic methods of determining the COR based on 2D X-ray imaging—Ranawat’s method. First, pelvic height must be determined by drawing two horizontal lines: the first at the level of the iliac crests, the second at the tubercles of the ischium. A right-angled triangle is then drawn, starting from a point located on average 5 mm laterally from the intersection of the Kohler and Shenton lines. The length of the side of the triangle is one fifth of pelvic height. The COR is in the half of the hypotenuse [13].

**Figure 2 diagnostics-13-02516-f002:**
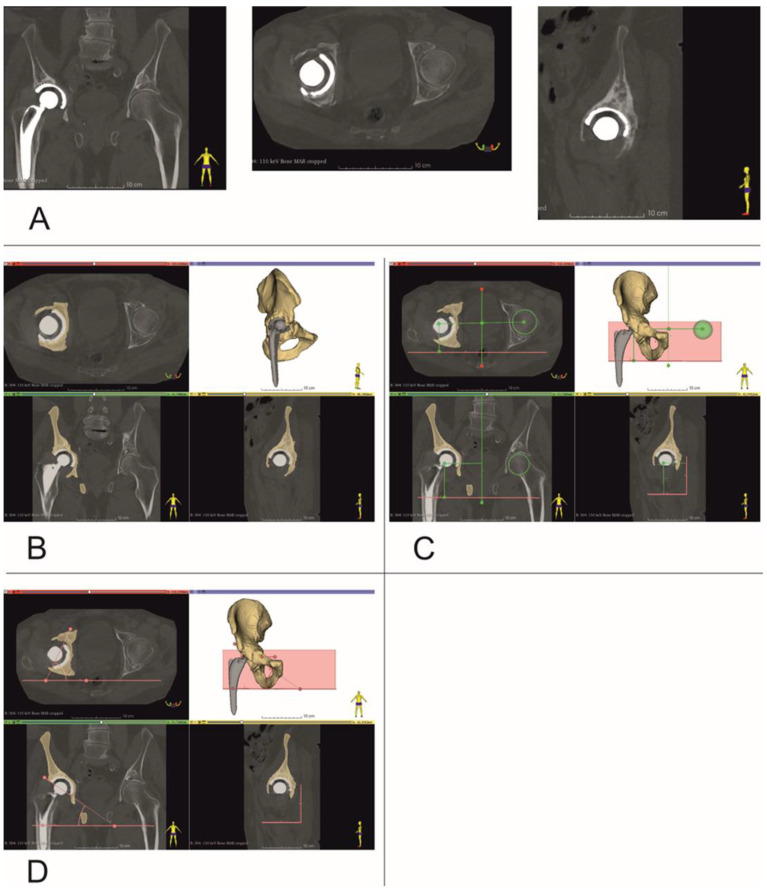
Establishing center of rotation, inclination and anteversion angles (Patient no. 10). (**A**) Coronal, Axial and Sagittal views, (**B**) bone tissue segmentation, and virtual 3D models of pelvis, (**C**) determining center of rotation, (**D**) inclination and anteversion angles.

**Figure 3 diagnostics-13-02516-f003:**
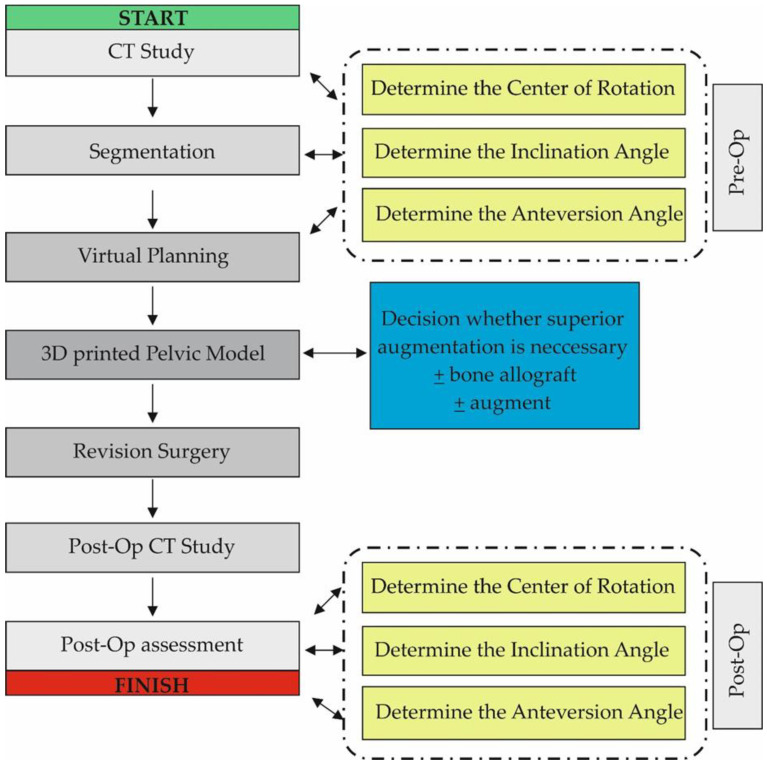
Diagram showing the workflow used in the study. The Post-Op Assessment stage was similar to the Pre-Op stage and consisted of CT data analysis, Segmentation and Virtual Planning.

**Figure 4 diagnostics-13-02516-f004:**
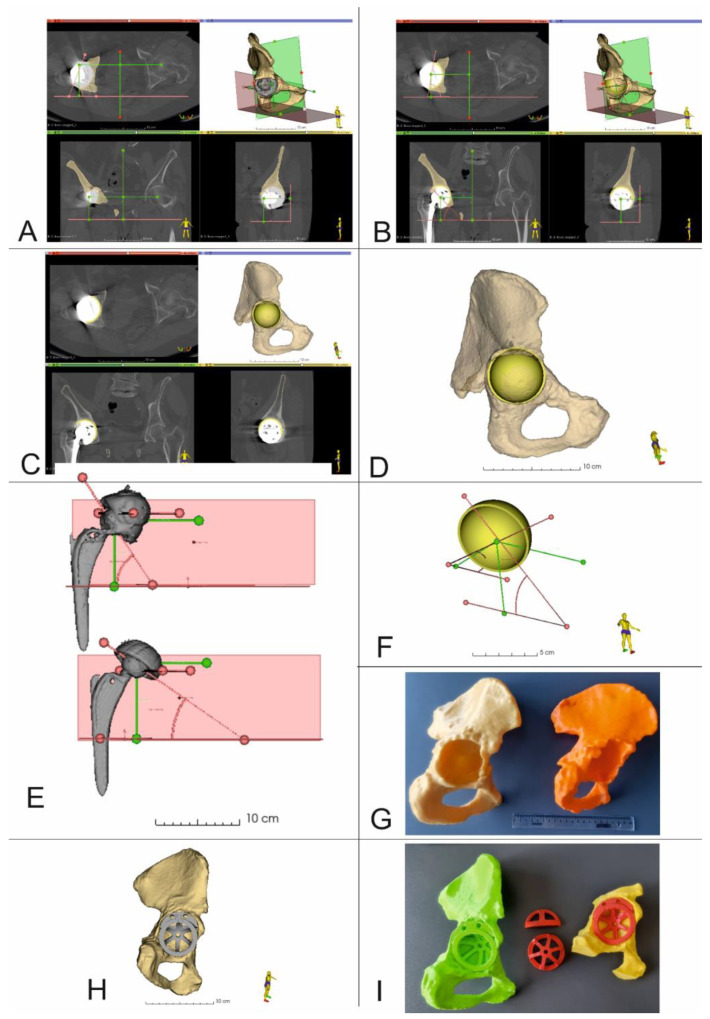
3D Virtual planning and comparison of acetabular cup positions before and after the procedure (Patient 5 and Patient 10). (**A**–**D**,**F**) 3D reconstruction of the pelvis with center of rotation shown together with planned position of the acetabular cup and screws (Patient no. 5), (**E**) Comparison of the COR, inclination, and anteversion angles before and after surgery (Patient no. 10), (**G**) 3D printed models of the pelvis. Yellow model *up and out* defect, orange model *up and in* defect, (**H**) 3D Virtual planning of acetabular cup and augment positions, (**I**) 3D printed models of the pelvis, acetabular cup and augment in the planned position.

**Figure 5 diagnostics-13-02516-f005:**
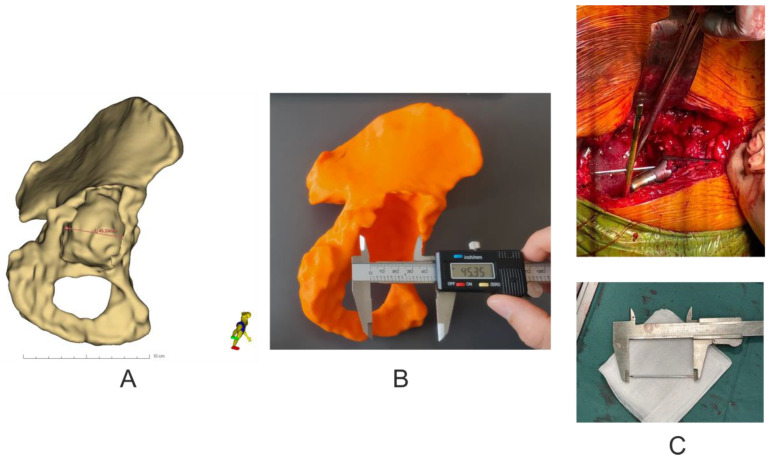
Assessment of the accuracy of making pelvic models. (**A**) Virtual measurement of the size of the pelvic bone defect in the axial plane. (**B**) Visual measurement of the size of the pelvic bone defect in the 3D model in the axial plane. (**C**) Measurement of the size of the pelvic bone defect using K-wire during revision surgery.

**Table 1 diagnostics-13-02516-t001:** CT scan parameters [23].

Computed Tomography Scanning Protocol	
Type of study	Single Phase, Non-contrast Enhanced Computed Tomography
Region	Pelvis
Patient position	Supine, lower limbs positioned symetrically
Image reconstruction	Reconstruction algorithms (Kernels): GE: Standard, Bone, Bone Plus; Siemens: B30f, B30s, B60s, B70s;
Slice Thickness	≤1.0 mm (isometric voxel)
Resolution	512 × 512
Pitch	≤1.0
Gantry Tilt Angle	0°
Data Format	Uncompressed DICOM files

**Table 2 diagnostics-13-02516-t002:** 3D printing parameters [23].

3D Printing Parameters	
Printing temperature	205 °C
Build plate temperature	60 °C
Layer high	0.2 mm
Infill	60%
Support material	Polyvinyl alcohol (PVA) (Ultimaker B.V, Utrecht, The Netherlands)

## Data Availability

Not applicable.

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
