# Peer review of "Optimization of Revision Hip Arthroplasty Workflow by Means of Detailed Pre-Surgical Planning Using Computed Tomography Data, Open-Source Software and Three-Dimensional-Printed Models"

_diagnostics, 2023, doi:10.3390/diagnostics13152516_

Round 1

Reviewer 1 Report

This research demonstrates a method to plan revision hip arthroplasty procedures, using open-source software in conjunction with lost-cost Fused Filament Fabrication 3D printers and anatomical models.

The abstract is well structured.

In the introduction, the research context is presented, and at the end of the paragraph, the objective of the study is stated. In line 101 it is not clear what "lost-cost FFF" means.

The materials and methods section clearly describes the research methodology followed, starting from clinical information, the technical aspects of determining dimensional parameters, virtual presurgical planning, 3D printing and statistical analysis.

The results paragraph is clearly and meticulously written.

The discussion section analyzes the results of the study in relation to other research presented in the literature. At the end of the paragraph, the limitations of the study are presented.

However, the discussion section can be expanded by comparing the results with other results obtained by using 3D technologies in traumatic orthopedic surgery, for example:

Moldovan F, Gligor A, Bataga T. Structured Integration and Alignment Algorithm: A Tool for Personalized Surgical Treatment of Tibial Plateau Fractures. J Pers Med. 2021 Mar 10;11(3):190. doi: 10.3390/jpm11030190.

The conclusions section is concise, and the bibliography is adequate and properly written.

There are minor editing errors in lines 80, 92.

I suggest reversing the order of figures 2 and 3 so that they are mentioned in the text in ascending order.

Figure 2 should be written with characters closer to those in the text.

Author Response

Detailed response to the Reviewer’s comments:

Manuscript ID: diagnostics-2476253

Title: Optimization of revision hip arthroplasty workflow by means of detailed pre-surgical planning using CT data, open-source software and 3D printed models.

Authors: Krzysztof Andrzejewski, Marcin Domżalski, Piotr Komorowski, Jan Poszepczyński, Bożena Rokita, Marcin Elgalal*

First of all, we would like to kindly thank the Reviewers for their effort and detailed comments, which we believe will help improve our manuscript. We have accepted almost all of the comments as well as made changes to the manuscripts in line with the reviewer suggestions.

Please find below our responses to the reviewer comments

Reviewer 1

Comments:

In the introduction, the research context is presented, and at the end of the paragraph, the objective of the study is stated. In line 101 it is not clear what "lost-cost FFF" means.

This is a mistake it should be “low-cost FFF”. This has been corrected. Furthermore, the text has been slightly modified in order to clarify this statement, as shown below:

The aim of this study was to demonstrate a practical and straightforward method to plan revision hip arthroplasty procedures, using open-source software in conjunction with lost low-cost FFF 3D printers and anatomical models The aim of this study was to demonstrate a practical and straightforward method to plan revision hip arthroplasty procedures, using open-source software in conjunction low-cost desktop FFF 3D printers that can produce significantly cheaper anatomical models [20], compared to other additive manufacturing technologies such as SLA, PolyJet or FDM.

The aim of this study was to demonstrate a practical and straightforward method to plan revision hip arthroplasty procedures, using open-source software in conjunction with low-cost desktop FFF 3D printers that can produce significantly cheaper anatomical models [20], compared to other additive manufacturing technologies such as SLA, PolyJet or FDM.

However, the discussion section can be expanded by comparing the results with other results obtained by using 3D technologies in traumatic orthopedic surgery, for example:

Moldovan F, Gligor A, Bataga T. Structured Integration and Alignment Algorithm: A Tool for Personalized Surgical Treatment of Tibial Plateau Fractures. J Pers Med. 2021 Mar 10;11(3):190. doi: 10.3390/jpm11030190.

According to the Reviewer’s suggestions the appropriate changes have been made to the Discussion.

Please find below the modified fragments of the manuscript.

Unfortunately, our study has two main limitations, the first and most important being the small number of cases analyzed. As a result of the small study population, it was difficult to clearly determine a homogeneous group of patients and perform a more thorough statistical analysis.  The second limitation  is the quality of the 3D printed models used in the study and, above all, accurate segmentation of pre and post-op CT data. Detailed 3D models of the revision site that are a result of meticulous segmentation of bone structures have a significant impact on a surgeon’s ability to assess the degree of tissue deformation and / or destruction of the acetabulum, especially in patients with a primary cemented acetabulum. During the process of learning to segment and 3D print such models, we observed a tendency, in particular among engineers, to “improve” 3D models using segmentation software smoothing algorithms. This resulted in less detailed models in which certain bone defects were difficult to discern and thus increased the risk of overlooking or underestimating the degree of bone tissue loss. With time, increased number of patient cases and after frequent consultations with other members of the team consisting of an orthopedic surgeon, radiologist and biomedical engineer, this tendency was eliminated. An additional method that could be possibly be used to assess 3D models of the operated acetabulum and aid in establishing the COR, is the application of an alignment technique and iterative closest point algorithms. Such solutions have been shown to be effective used in trauma patients and may be helpful in pre-surgical planning of RHA [44] Another method is the use of neural network (NN) and artificial intelligence (AI) in the process of bone tissue segmentation [38 45]. Wu et al. [39 46] showed that the use of new neural networks (e.g. CNN-CMG Net) increases the accuracy of bone tissue segmentation while reducing the time required to create 3D models of such anatomical structures, which clearly indicates a potential research direction to optimize the presurgical planning process. However, it should be emphasized that these promising results focused on healthy bone tissue. The authors clearly indicated that their work did not concern segmentation of bone structures in which metal implants were present. Nevertheless, it seems that the current development of information technology based on AI is a very promising and it is only a matter of time when surgical planning for THA and RHA will be based on neural networks and possibly simplified tissue segmentation methods [40-42 47-49].

There are minor editing errors in lines 80, 92.

These errors have been corrected.

I suggest reversing the order of figures 2 and 3 so that they are mentioned in the text in ascending order.

The order of figures 2 and 3 has been reversed.

Figure 2 should be written with characters closer to those in the text.

The character font and size has been modified and is now the same as that of the main text.

Figure 3. Diagram showing the workflow used in the study. The Post-Op Assessment stage was similar to the Pre-Op stage and consisted of CT data analysis, Segmentation and Virtual Planning. Establishing center of rotation, inclination and anteversion angles (Patient no. 10). (A) Coronal, Axial and Sagittal views, (B) bone tissue segmentation, and virtual 3D models of pelvis, (C) determining center of rotation, (D) inclination and anteversion angles.

Reviewer 2 Report

・A detailed explanation is required as to why it is effective to use a 3D printer and preoperative CT planning together. Background should be provided that using each technique alone is insufficient.

・Insufficient description of surgical technique. Is X-ray fluoroscopy used?

・There are no controls in this study. In order to prove the hypothesis of this study, it is necessary to compare with cases where each procedure was used alone.

I think that there is no problem in general.

Author Response

Detailed response to the Reviewer’s comments:

Manuscript ID: diagnostics-2476253

Title: Optimization of revision hip arthroplasty workflow by means of detailed pre-surgical planning using CT data, open-source software and 3D printed models.

Authors: Krzysztof Andrzejewski, Marcin Domżalski, Piotr Komorowski, Jan Poszepczyński, Bożena Rokita, Marcin Elgalal*

First of all, we would like to kindly thank the Reviewers for their effort and detailed comments, which we believe will help improve our manuscript. We have accepted almost all of the comments as well as made changes to the manuscripts in line with the Reviewer suggestions.

Please find below our responses to the Reviewer comments

Reviewer 2

Comments:

A detailed explanation is required as to why it is effective to use a 3D printer and preoperative CT planning together. Background should be provided that using each technique alone is insufficient.

According to the Reviewer’s suggestions the appropriate changes have been made to the Introduction.

Please find below the modified fragments of the manuscript.

A good way to solve this problem is the use volumetric medical imaging, reconstruction software, and virtual planning together with 3D printing and anatomical models [1]. By combing these diverse techniques into one process, it is possible to take advantage of the recent developments in these technologies and use them in the field of medicine, in particular, surgery. Detailed medical imaging in the form of Computed Tomography or Magnetic Resonance Imaging (MRI) together with appropriate software, enables surgeons to perform virtual planning of complicated operations. This process can be further enhanced through the use of patient specific anatomical models created using additive manufacturing methods, which in recent times have become much more accessible with the rapid development of budget desktop 3D printers. With such models, surgeons are able to physical assess the size and shape of the affected pelvis, which in turn makes it easier to classify the type of defect present, plan the use of metal augments or allografts (Figure 2) and any necessary bone screws. Furthermore, senior doctors can use these anatomical models as a teaching tool [20] when training junior colleagues to perform complex procedures, which is an essential part of the learning process for future surgeons. Finally, 3D anatomical models have been shown to be effective devices for patient education [21] due to the fact that they can visualize their specific conditions. Despite these numerous advantages, to date, these techniques have not been commonly used in medicine, especially in the everyday clinical practice of orthopedics. Despite the numerous developments of software used in the analysis and reconstruction of different medical images (i.e., CT, MRI) as well as advances in additive technologies, to date, these techniques are not commonly used in medicine, especially in the everyday clinical practice of orthopedics.

[20] Divya Mehrotra, A.F. Markus. Emerging simulation technologies in global craniofacial surgical training. Journal of Oral Biology and Craniofacial Research, Volume 11, Issue 4, 2021,. https://doi.org/10.1016/j.jobcr.2021.06.002

[21] Yuan-dong Zhuang, Mao-chao Zhou, Shi-chao Liu, Jian-feng Wu, Rui Wang, Chun-mei Chen. Effectiveness of personalized 3D printed models for patient education in degenerative lumbar disease. Patient Education and Counseling, Volume 102, Issue 10, 2019, https://doi.org/10.1016/j.pec.2019.05.006

Insufficient description of surgical technique. Is X-ray fluoroscopy used?

According to the Reviewer’s suggestions the appropriate changes have been made to the Clinical information.

Please find below the modified fragments of the manuscript.

Clinical information

Between January 2021 and July 2022, a total of 10 patients (4 male and 6 female), aged 50 – 65 years with an average age of 60 years, who underwent revision HA were included in the study. The study population, although small, was homogeneous with the exception of one patient (single case of up and out type deformity). All patients had undergone primary unilateral THA with cemented acetabular cups that had become loosened, Paprosky type II and IIIA acetabular defects, vertical upward migration > 3cm and intact Kohler's line. For each patient a posterior approach was carried out using the Kocher-Langenbeck - lateral position. In order to prevent injury to the sciatic nerve, the ipsilateral knee was kept in a flexed position during the procedure. Primary endoprosthesis stem removal was not required in any of the cases. Standard antibiotic prophylaxis with Cefazolin was administered. Intraoperative X-ray fluoroscopy was not used, however radiographic imaging was performed immediately after skin closure. All patients were operated on by the same surgical team and orthopedic department. For all patients, the same implant system was used, i.e., Trabecular Metal Acetabular Revision System (Zimmer Biomet). In case of insufficient contact of the revision acetabulum with the pelvic bone, an allograft was used. This study was approved by the Ethics Committee on Human Research of the Medical University of Lodz, No. RNN/126/22/KE.

Intraoperative X-ray fluoroscopy was not used for any of the procedures

There are no controls in this study. In order to prove the hypothesis of this study, it is necessary to compare with cases where each procedure was used alone.

The main objective of our study was to demonstrate the use of open-source software and patient specific anatomical models in planning revision hip arthroplasty. The center of rotation for the revision acetabulum is calculated on the basis of CT imaging, using the software platform 3D Slicer. 3D printed models are then used as an additional tool for surgical pre-planning and simulation. It would be possible to perform this procedure without using anatomical models i.e., separately, as suggested by the reviewer. However, it would be impossible to do the same using just 3D printed models, due to the fact that without any reference planes or a coordinate system, it would not be possible to accurately position any objects in a manner that would be reproducible. Unless some form of sophisticated positioning equipment was used, any assessment of the COR position would only be subjective, without the possibility of acquiring any objective parameters. To our knowledge, in the literature, physical 3D models are usually used in conjunction with CT data and are typically created on the basis of these images. For these reasons the authors believe that comparing cases in which these two techniques are used separately would be challenging. Moreover, as mentioned above our main objective was to demonstrate the use of relatively easily available equipment and software for planning revision HA.

Reviewer 3 Report

Dear authors, i had the opportunity to review your work and manuscript. I appreciate your effort; however, I would like to add some comments.

-The authors build their study on the fact that they will use a method for preoperative planning to be effective and of low cost for efficiency and accuracy; this could be proved by the numbers; however, for the issue of cost, how did the authors assume that this is of a low cost? compared to what? did they calculate the cost of the CT, the pelvic model printing, and the time consumed during the process? 

-what was the main role of printing a pelvic model, I noticed that the authors only measured the size of the defect in this model and compared it with the intraoperative value, which could be obtained from the CT images itself without the need to print a model.

-Line 42 in the abstract it should be "defects", not deformities

-Line 123, and 124: figure 3 is coming before Figure 2??

-Line 132, reporting that the intact side is "uninjured" and the revision side "injured" is not appropriate; this is not a trauma scenario; please correct throughout the whole manuscript.

-in the planing process, the authors relied on the fact that the contralateral side is normal, but how this will be done if the other side is deformed or both sides need revision? Will the technique the authors described applies only to unilateral cases with a normal contralateral side?

-what are the references for the inclination and anteversion? did the authors consider the Lewinnek safe zone a s the reference or they tailored these numbers according to each patient?

-the authors have to mention the types of defect they have dealt with, especially in this small series, as I noticed that most of the discussion was about in-and-up defects with an intact Kohlers line, at which step the authors determined the need for a superior augmentation (as they reported that they used allografts in some cases)? was this determined during planning or intraoperatively? It is preferable to demonstrate such a step on the printed pelvic model.

-the results section needs to be simplified, and do not repeat the data mentioned in the tables again in the text and vice versa.

Minimal language polishing is needed

Author Response

Detailed response to the Reviewer’s comments:

Manuscript ID: diagnostics-2476253

Title: Optimization of revision hip arthroplasty workflow by means of detailed pre-surgical planning using CT data, open-source software and 3D printed models.

Authors: Krzysztof Andrzejewski, Marcin Domżalski, Piotr Komorowski, Jan Poszepczyński, Bożena Rokita, Marcin Elgalal*

First of all, we would like to kindly thank the Reviewers for their effort and detailed comments, which we believe will help improve our manuscript. We have accepted almost all of the comments as well as made changes to the manuscripts in line with the Reviewer suggestions.

Please find below our responses to the Reviewer comments

Reviewer 3

Comments:

The authors build their study on the fact that they will use a method for preoperative planning to be effective and of low cost for efficiency and accuracy; this could be proved by the numbers; however, for the issue of cost, how did the authors assume that this is of a low cost? compared to what? did they calculate the cost of the CT, the pelvic model printing, and the time consumed during the process?

The main objective of our study was to plan revision HA using CT data, reconstruction software and 3D printed models. CT scans were performed for patients who were being treated by the department of Orthopedics. Regardless of how any surgical preplanning would be carried out for these patients, they would each need to have a CT scan performed before any decisions regarding their surgical procedures could be established. Therefore, we did not take into consideration the cost of CT scanning.

In order to view and reconstruct acquired CT data, appropriate software has to be used, that is also capable of converting these images in printable 3D models. There are several commercial software packages available on the market that can perform such tasks, however they cost in the range of $ 7 500 – $30 - 40 000. For this reason, we choose an open-source package that was free to use.

With regards to 3D printing the anatomical models we choose a desktop printer that uses FFF (Fused Filament Fabrication) technology. There are numerous 3D printers on the market that use all different types of printing technologies i.e., SLA, FDM, PolyJet. The cost of these printers, consumables and printed models differs significantly. FFF technology originated from the open-source project RepRap that led to the rapid development of numerous desktop 3D printers around the world. The cost of these printers and more importantly the cost of printing anatomical models using this technology has been shown to be significantly lower than other techniques.

We therefore used free open-source software together with the cheapest 3D printing technology available on the market and for these reasons we described this as a low-cost method

The authors did not take into consideration the time / man hours required to perform this planning.

Reference:

[1] Chen JV, Dang ABC, Dang A. Comparing cost and print time estimates for six commercially-available 3D printers obtained through slicing software for clinically relevant anatomical models. 3D Print Med. 2021 Jan 6;7(1):1. doi: 10.1186/s41205-020-00091-4. PMID: 33404847; PMCID: PMC7786189

[2] https://www.materialise.com/en/healthcare/mimics-innovation-suite/mimics

[3] https://oqton.com/d2p/

[4] Aljibe, M.S.O., Bundoc, R.C., Acos , R.L.C., Adolfo, J.A.L., Adorna, C.G., Agner, A.D.G., Alejandre, A.C.T., Alfonso, P.G.I., Alip III, A.B.L., Almanza, J.R.T., Alonday , S.P.H., Amilhasan, F.S. and Anarna, K.S. 2022. Low-cost 3D Modeling Software for Generating Patient-specific Drill Guide Templates for Cervical Pedicle Screw Insertion: An In Vitro Study. Acta Medica Philippina. 56, 20 (Dec. 2022). DOI:https://doi.org/10.47895/amp.v56i20.6571.

what was the main role of printing a pelvic model, I noticed that the authors only measured the size of the defect in this model and compared it with the intraoperative value, which could be obtained from the CT images itself without the need to print a model.

According to the authors the main role of printing a pelvic model is:

…….With such models, surgeons are able to physical assess the size and shape of the affected pelvis, which in turn makes it easier to classify the type of defect present, plan the use of metal augments or allografts (Figure 2) and any necessary bone screws. Furthermore, senior doctors can use these anatomical models as a teaching tool [20] when training junior colleagues to perform complex procedures, which is an essential part of the learning process for future surgeons. Finally, 3D anatomical models have been shown to be effective devices for patient education [21]….

[20] Divya Mehrotra, A.F. Markus. Emerging simulation technologies in global craniofacial surgical training. Journal of Oral Biology and Craniofacial Research, Volume 11, Issue 4, 2021,. https://doi.org/10.1016/j.jobcr.2021.06.002.

[21] Yuan-dong Zhuang, Mao-chao Zhou, Shi-chao Liu, Jian-feng Wu, Rui Wang, Chun-mei Chen. Effectiveness of personalized 3D printed models for patient education in degenerative lumbar disease. Patient Education and Counseling, Volume 102, Issue 10, 2019, https://doi.org/10.1016/j.pec.2019.05.006

Measurements of the defects on the anatomical models were carried out as an additional precaution to assess the accuracy of the segmentation process as well as the 3D printed models.

There are several works in the literature that focus on these issues, one such study has been cited in the manuscript.

Eventually, according to the Reviewer’s suggestions the appropriate changes have been made to the Introduction.

Please find below the modified fragments of the manuscript.

A good way to solve this problem is the use volumetric medical imaging, reconstruction software, and virtual planning together with 3D printing and anatomical models [1]. By combing these diverse techniques into one process, it is possible to take advantage of the recent developments in these technologies and use them in the field of medicine, in particular, surgery. Detailed medical imaging in the form of Computed Tomography (CT) or Magnetic Resonance Imaging together with appropriate software, enables surgeons to perform virtual planning of complicated operations. This process can be further enhanced through the use of patient specific anatomical models created using additive manufacturing methods, which in recent times have become much more accessible with the rapid development of budget desktop 3D printers. With such models, surgeons are able to physical assess the size and shape of the affected pelvis, which in turn makes it easier to classify the type of defect present, plan the use of metal augments or allografts (Figure 2) and any necessary bone screws. Furthermore, senior doctors can use these anatomical models as a teaching tool [20] when training junior colleagues to perform complex procedures, which is an essential part of the learning process for future surgeons. Finally, 3D anatomical models have been shown to be effective devices for patient education [21] due to the fact that they can visualize their specific conditions. Despite these numerous advantages, to date, these techniques have not been commonly used in medicine, especially in the everyday clinical practice of orthopedics. Despite the numerous developments of software used in the analysis and reconstruction of different medical images (i.e., CT, MRI) as well as advances in additive technologies, to date, these techniques are not commonly used in medicine, especially in the everyday clinical practice of orthopedics.

Line 42 in the abstract it should be "defects", not deformities

According to the Reviewer’s suggestions the appropriate changes have been made to the manuscript.

Please find below the modified fragments of the manuscript.

3D reconstruction software, together with low-cost anatomical models are very effective tools for pre-surgical planning, which have great potential use in orthopedic surgery, particularly RHA. In up and in and up and out type deformities defects, it is essential to establish a new COR and to identify three support points within the revision acetabulum in order to correctly position acetabular cups.

The study population, although small, was homogeneous with the exception of one patient (single case of up and out type deformity defect).

Figure 4. 3D Virtual planning and comparison of acetabular cup positions before and after the procedure (Patient 5 and Patient 10). (A), (B), (C), (D), (F) 3D reconstruction of the pelvis with center of rotation shown together with planned position of the acetabular cup and screws (Patient no. 5), (E) Comparison of the COR, inclination, and anteversion angles before and after surgery (Patient no. 10), (G) 3D printed models of the pelvis. Yellow model up and out deformity defect, orange model up and in deformity defect.

Furthermore, the use of these techniques is not limited to pelvic surgery, but could be applied to spinal surgery for the purpose of planning pedicle insertion and osteotomy procedures in cases of spinal deformity defect [35 39], craniofacial surgery to prepare custom-made titanium mandibular protheses [36 40], as well as cancer treatment and the use of MRI-based three-dimensional models e.g. planning of radical prostatectomy [37 41].

In up and in and up and out type deformities defects of the, it is essential to identify the three support points of the revision acetabulum. Identification of acetabular roof, ischium, pubic bone, and acetabular wall defects is possible thanks to 3D printing and CT image segmentation. Determining the postoperative center of rotation based on follow-up CT data is essential for assessing the success of the operation.

Line 123, and 124: figure 3 is coming before Figure 2?

According to the Reviewer’s suggestions the appropriate changes have been made to the order of figures

Please find below the modified fragments of the manuscript:

CT data were saved as DICOM files (Figure 2 3 (A)). Segmentation of bone structures (Figure 2 3 (B)) was carried out using the free open-source software platform 3D Slicer 5.03 [18 22]. The center of rotation, inclination, and anteversion angles were established using the following workflow (Figure 3 2).

Figure 2. Establishing center of rotation, inclination and anteversion angles (Patient no. 10). (A) Coronal, Axial and Sagittal views, (B) bone tissue segmentation, and virtual 3D models of pelvis, (C) determining center of rotation, (D) inclination and anteversion angles. Diagram showing the workflow used in the study.

Figure 3. Diagram showing the workflow used in the study. The Post-Op Assessment stage was similar to the Pre-Op stage and consisted of CT data analysis, Segmentation and Virtual Planning. Establishing center of rotation, inclination and anteversion angles (Patient no. 10). (A) Coronal, Axial and Sagittal views, (B) bone tissue segmentation, and virtual 3D models of pelvis, (C) determining center of rotation, (D) inclination and anteversion angles.

Line 132, reporting that the intact side is "uninjured" and the revision side "injured" is not appropriate; this is not a trauma scenario; please correct throughout the whole manuscript.

According to the Reviewer’s suggestions the appropriate changes have been made to the Materials and Methods “Determining Center of Rotation, Inclination and Anteversion angles”

Please find below the modified fragments of the manuscript:

Step 2, a sphere was drawn with dimensions corresponding to the uninjured intact side femur head and its center aligned with the center of the femur head. The distance, at right angles, of the center of this sphere relative to all three planes created in Step 1, was then measured.

Step 3, a second sphere with identical dimensions was drawn on the contralateral side, i.e., the injured revision side, revision acetabulum. This sphere was positioned with its center at identical distances to the three planes, created in Step 1, to those of the sphere on the uninjured intact side that was aligned with the femur head. Thus, a new center of rotation was established, with XYZ coordinates based on the COR of the contralateral uninjured intact side femur head (Figure 2 3 (C)).

in the planning process, the authors relied on the fact that the contralateral side is normal, but how this will be done if the other side is deformed or both sides need revision? Will the technique the authors described applies only to unilateral cases with a normal contralateral side?

The current Technical Note is the first work in a series of studies related to pre-surgical planning of RHA using open-source software and anatomical models built on a low-cost 3D printer. This study focuses on patients with unilateral Paprosky type II defects, due to the fact that such cases are less complex in nature. Thus, we have been able to develop our workflow and methodology for creating 3D models, as well as establishing COR together with inclination and anteversion angles. On the basis of this experience we are currently working on applying these techniques to more complex cases such as unilateral and bilateral Paprosky type III defects.

what are the references for the inclination and anteversion? did the authors consider the Lewinnek safe zone a s the reference or they tailored these numbers according to each patient?

The reference for the inclination angle was 45 deg.

The reference for the anteversion angle was 14 deg.

We did not directly use the Lewinnek safe zone [1] to analyze our results. However, the values we achieved for the inclination and anteversion angles were well within the "safe zone" ie., inclination angle 30 to 50 deg., and anterversion angle 5 deg. to 25 deg.

Reference:

[1] Bosker, B.H., Verheyen, C.C.P.M., Horstmann, W.G. et al. Poor accuracy of freehand cup positioning during total hip arthroplasty. Arch Orthop Trauma Surg 127, 375–379 (2007). https://doi.org/10.1007/s00402-007-0294-y

the authors have to mention the types of defect they have dealt with, especially in this small series, as I noticed that most of the discussion was about in-and-up defects with an intact Kohlers line, at which step the authors determined the need for a superior augmentation (as they reported that they used allografts in some cases)? was this determined during planning or intraoperatively? It is preferable to demonstrate such a step on the printed pelvic model.

While preparing responses to the reviewer comments, we noticed that we made a serious mistake regarding the types of defects discussed in our work. The majority of the patients in our study group were Paprosky type II defects and not type III as described. In the present study we only had one patient with a Paprosky type III A defect. We have started work on a second study that focuses on type III A and III B defects, and we accidently described the patients in this study inaccurately. We would like to sincerely apologize for this unfortunate mistake.

The decision to use superior augmentation or an allograft was initially made at the pre-surgical planning stage. We analyzed the extent of the defect within the affected side, chose an appropriate revision socket size and determined the need for superior augmentation in order ensure implant stability (Figure A). A 3D print of the damaged pelvis together with a model of the acetabular cup and a potential augment were prepared (Figure B). The final decision regarding acetabular cup size and whether to use an augment was made intraoperatively by the attending surgeon.

Please find below the modified fragments of the manuscript.

Abstract: Background. In Revision Hip Arthroplasty (RHA), establishing the center of rotation (COR) can be technically challenging due to the acetabular bone destruction that is usually present, particularly severe cases such as Paprosky type II and III defects. The aim of this study was to demonstrate the use of open-source, medical image reconstruction software and low-cost 3D anatomical models in pre-surgical planning of RHA.

Materials and Methods

Clinical information

Between January 2021 and July 2022, a total of 10 patients (4 male and 6 female), aged 50 – 65 years with an average age of 60 years, who underwent revision HA were included in the study. The study population, although small, was homogeneous with the exception of one patient (single case of up and out type deformity defect). All patients had undergone primary unilateral THA with cemented acetabular cups that had become loosened, Paprosky type II and IIIA (i.e., vertical upward migration > 3cm and intact Kohler's line) acetabular defects. For each patient a posterior approach was carried out using the Kocher-Langenbeck - lateral position. In order to prevent injury to the sciatic nerve, the ipsilateral knee was kept in a flexed position during the procedure. Primary endoprosthesis stem removal was required in any of the cases. Standard antibiotic prophylaxis with Cefazolin was administered. Intraoperative X-ray fluoroscopy was not used, however radiographic imaging was performed immediately after skin closure. All patients were operated on by the same surgical team and orthopedic department. For all patients, the same implant system was used, i.e. Trabecular Metal Acetabular Revision System (Zimmer Biomet). In case of insufficient contact of the revision acetabulum with the pelvic bone, an allograft was used. This study was approved by the Ethics Committee on Human Research of the Medical University of Lodz, No. RNN/126/22/KE.

According to the Reviewer’s suggestions the appropriate changes have been made to the Table 3 and we added information about the types of defect (Paprosky classification).

Please find below the modified fragments of the manuscript.

Table 63. Accuracy analysis of 3D models and information about types of defect 

(Paprosky classification) (mean ± SD, n=10).

Information about types of defect
(Paprosky classification)

3D virtual models
measurement/

3D model measurement

results
mm (% difference)

3D model measurement/

pelvis of patient
measurement

results
mm (% difference)

3D virtual model measurement/

pelvis of patient measurement

results mm
(% difference)

Patient 1 (Paprosky IIB)

-0.15/99.66

0.25/100.56

0.10/100.23

Patient 2 (Paprosky IIC)

-0.04/99.92

0.10/100.19

0.06/100.12

Patient 3 (Paprosky IIB)

-0.11/99.80

0.16/100.30

0.05/100.09

Patient 4 (Paprosky IIA)

-0.70/98.59

1.50/103.12

0.80/101.67

Patient 5 (Paprosky IIB)

-0.21/99.63

0.81/101.45

0.60/101.07

Patient 6 (Paprosky IIB)

0.05/100.09

0.51/100.93

0.56/101.02

Patient 7 (Paprosky IIC)

0.21/100.36

0.40/99.32

-0.19/99.67

Patient 8 (Paprosky IIA)

-0.05/99.91

0.14/99.76

-0.19/99.67

Patient 9 (Paprosky IIB)

-0.15/99.74

0.04/99.93

-0.19/99.68

Patient 10 (Paprosky IIIA)

-0.03/99.94

0.04/99.91

-0.07/99.85

mean ± SD

-0.12 ± 0.24/99.76 ± 0.47

0.27 ± 0.55/100.55 ± 1.09

0.15 ± 0.37/100.31 ± 0.70

as I noticed that most of the discussion was about in-and-up defects with an intact Kohlers line, at which step the authors determined the need for a superior augmentation (as they reported that they used allografts in some cases)? was this determined during planning or intraoperatively? It is preferable to demonstrate such a step on the printed pelvic model.

While preparing responses to the reviewer comments, we noticed that we made a serious mistake regarding the types of defects discussed in our work. The majority of the patients in our study group were Paprosky type II defects and not type III as described. In the present study we only had one patient with a Paprosky type III A defect. We have started work on a second study that focuses on type III A and III B defects, and we accidently described the patients in this study inaccurately. We would like to sincerely apologize for this unfortunate mistake.

For the patient with the type III A defect, the decision to use superior augmentation or an allograft was initially made at the pre-surgical planning stage. We analyzed the extent of the defect within the affected side, chose an appropriate revision socket size and determined the need for superior augmentation in order ensure implant stability (Figure A). A 3D print of the damaged pelvis together with a model of the acetabular cup and a potential augment were prepared (Figure B). The final decision regarding acetabular cup size and whether to use an augment was made intraoperatively by the attending surgeon.

A

B

For the remaining cases i.e., type II defects two bone screws were used to stabilize the acetabular cups.

the results section needs to be simplified, and do not repeat the data mentioned in the tables again in the text and vice versa

According to the Reviewer’s suggestions the appropriate changes have been made to the Results.

We have transferred tables from the manuscript to supplementary materials.

Please find below the modified fragments of the manuscript and supplementary materials:

Table 3. Centre of rotation position (mean ± SD, n=10).

Preoperative distance

[mm]

3D planning distance

[mm]

Postoperative distance

[mm]

X-axis
(Coronal plane)

86.3 ± 8.7

86.0 ± 7.8

87.5 ± 8.6

Y-axis
(Coronal plane)

71.7 ± 9.6

71.3 ± 11.8

66.6 ± 10.5

Z-axis
(Axial plane)

52.2 ± 4.8

51.8 ± 2.3

53.7 ± 4.5

Table 4. Anteversion and angle of inclination values (mean ± SD, n=10).

Parameters

Preoperative

[deg.]

3D planning

[deg.]

Postoperative

[deg.]

Inclination angle

55.7 ± 15.4

47.7 ± 5.6

51.6 ± 5.3

Anteversion angle

21.1 ± 9.7

17.3 ± 4.1

15.4 ± 5.2

Table 5. Correlation of presurgical 3D planning and post-operative COR position measurements, acetabular and inclination angles (mean ± SD, n=10, p < 0.05, *- statistical significant, ns- no significant).

3D planning

results
mm / deg.

Postoperative

results
mm / deg.

Pearson r

p value

X-axis
(Coronal plane)

86.0 ± 7.8

87.5 ± 8.6

0.6720 (ns)

0.0982

Y-axis
(Coronal plane)

71.3 ± 11.8

66.6 ± 10.5

0.9438 *

0.0014

Z-axis
(Axial plane)

51.8 ± 2.3

53.7 ± 4.5

0.8829 *

0.0084

Inclination angle

47.7 ± 5.6

51.6 ± 5.3

-0.4042 (ns)

0.3685

Anteversion angle

17.3 ± 4.1

15.4 ± 5.2

-0.4782 (ns)

0.2777

Table 63. Accuracy analysis of 3D models (mean ± SD, n=10).

Supplementary Materials

Optimization of revision hip arthroplasty workflow by means of detailed pre-surgical planning using CT data, open-source software and 3D printed models

Krzysztof Andrzejewski 1, Marcin Domżalski 1, Piotr Komorowski 3,
Jan Poszepczyński 1, Bożena Rokita 4 and Marcin Elgalal 2,*

1 Veteran’s Memorial Hospital Medical University of Lodz, Zeromskiego 113, 90-549 Lodz, Poland; [email protected]; [email protected]; [email protected]

2 Second Department of Radiology and Diagnostic Imaging, Medical University of Lodz, Pomorska 251, 92-213 Lodz, Poland; [email protected]

3 Division of Biophysics, Institute of Materials Science, Lodz University of Technology, Stefanowskiego 1/15, 90-924 Lodz, Poland; [email protected]

4 Institute of Applied Radiation Chemistry, Faculty of Chemistry, Lodz University of Technology, Wroblew-skiego 15, 93-590 Lodz, Poland; [email protected]

*Correspondence: [email protected]; Tel.: +48 42 201 42 06

Table S1. Centre of rotation position (mean ± SD, n=10).

Preoperative distance

[mm]

3D planning distance

[mm]

Postoperative distance

[mm]

X-axis
(Coronal plane)

86.3 ± 8.7

86.0 ± 7.8

87.5 ± 8.6

Y-axis
(Coronal plane)

71.7 ± 9.6

71.3 ± 11.8

66.6 ± 10.5

Z-axis
(Axial plane)

52.2 ± 4.8

51.8 ± 2.3

53.7 ± 4.5

Table S2. Anteversion and angle of inclination values (mean ± SD, n=10).

Parameters

Preoperative

[deg.]

3D planning

[deg.]

Postoperative

[deg.]

Inclination angle

55.7 ± 15.4

47.7 ± 5.6

51.6 ± 5.3

Anteversion angle

21.1 ± 9.7

17.3 ± 4.1

15.4 ± 5.2

Table S3. Correlation of presurgical 3D planning and post-operative COR position measurements, acetabular and inclination angles (mean ± SD, n=10, p < 0.05, *- statistical significant, ns- no significant).

3D planning

results
mm / deg.

Postoperative

results
mm / deg.

Pearson r

p value

X-axis
(Coronal plane)

86.0 ± 7.8

87.5 ± 8.6

0.6720 (ns)

0.0982

Y-axis
(Coronal plane)

71.3 ± 11.8

66.6 ± 10.5

0.9438 *

0.0014

Z-axis
(Axial plane)

51.8 ± 2.3

53.7 ± 4.5

0.8829 *

0.0084

Inclination angle

47.7 ± 5.6

51.6 ± 5.3

-0.4042 (ns)

0.3685

Anteversion angle

17.3 ± 4.1

15.4 ± 5.2

-0.4782 (ns)

0.2777

Reviewer 4 Report

Some comments given to authors for revision as follows:

1.      Line 16, abstract seems too long. Please make it more concise.

2.      Line 78, the Figure 1 encouraged to giving significant improvement since it is not looks so scientific.

3.      Line 54, can the authors justify the novel and significance in the present technical notes?

4.      In line58-59m since the authors have been used reff. [3] previously, please give additional or replace with another reference as follows: https://doi.org/10.3390/biomedicines11030951, https://doi.org/10.1038/s41598-023-30725-6, and https://doi.org/10.3390/su142013413

5.      Line 104, the patient involved is literally small. It is possibility to added?

6.      Line 115, this section is important in the technical note, please more explanation in this section.

7.      Line 120, nothing any parameter description? I believe there is other that not author mentioned.

8.      Line 127, for explaining the workflow, please try to give start-finish type with coloured step.

-

Author Response

Detailed response to the Reviewer’s comments:

Manuscript ID: diagnostics-2476253

Title: Optimization of revision hip arthroplasty workflow by means of detailed pre-surgical planning using CT data, open-source software and 3D printed models.

Authors: Krzysztof Andrzejewski, Marcin Domżalski, Piotr Komorowski, Jan Poszepczyński, Bożena Rokita, Marcin Elgalal*

First of all, we would like to kindly thank the Reviewers for their effort and detailed comments, which we believe will help improve our manuscript. We have accepted almost all of the comments as well as made changes to the manuscripts in line with the Reviewer suggestions.

Please find below our responses to the Reviewer comments

Reviewer 1

Comments:

Line 16, abstract seems too long. Please make it more concise.

According to the Reviewer’s suggestions the appropriate changes have been made to the Abstract.

Please find below the modified fragments of the manuscript:

Abstract: Background. In Revision Hip Arthroplasty (RHA), establishing the center of rotation (COR) can be technically challenging due to the acetabular bone destruction that is usually present, particularly severe cases such as Paprosky type III defects. The aim of this study was to demonstrate the use of open-source, medical image reconstruction software and low-cost 3D anatomical models in pre-surgical planning of RHA. Methods. Between January 2021 and July 2022,  A total of 10 patients, aged 50 – 65 years, underwent RHA and were included in the study. Computer Tomography (CT) scans were performed for all cases, before surgery and approximately 1 week after the procedure. The reconstruction of CT data, 3D virtual planning of the COR as well as positioning of acetabular cups, including their inclination and anteversion angles, was carried out using the free open source software platform 3D Slicer. In addition, anatomical models of the pelvis were built on a desktop 3D printer from polylactic acid (PLA). Preoperative and postoperative reconstructed imaging data were compared for each patient and the position of the acetabular cups as well as the COR were evaluated for each case. Data were compared using a two-sided paired T test and correlations between variables were determined using the Pearson correlation coefficient. Results. Analysis of the pre- and post-op center of rotation position data indicated statistically insignificant differences for the location of the COR on the X-axis (1.5 mm, t= 0.5741, p= 0.5868) with a fairly strong correlation of the results (r= -0.672, p= 0.0982). Whilst, for the location of the COR in the Y and Z-axes, there was statistical dependence (Y axis, 4.7 mm, t= 3.168 and p= 0.0194; Z axis, 1.9 mm, t= 1.887 and p= 0.1081). A strong correlation for both axes was also observed (Y and Z) (Y-axis, r= 0.9438 and p= 0.0014; Z-axis, r= 0.8829 and p= 0.0084). Analysis of inclination angle values showed a statistically insignificant difference between mean values (3.9 degrees, t = 1.111, p= 0.3092) and a moderate correlation was found between mean values (r= -0.4042, p= 0.3685). Analysis of the anteversion angle showed a statistically insignificant difference between mean values (1.9 degrees, t = 0.8671, p= 0.4192), while a moderate correlation between mean values was found (r= -0.4782, p= 0.2777). Conclusion. 3D reconstruction software, together with low-cost anatomical models are very effective tools for pre-surgical planning, which have great potential use in orthopedic surgery, particularly RHA. In up and in and up and out type deformities, it is essential to establish a new COR and to identify three support points within the revision acetabulum in order to correctly position acetabular cups. Assessment of the acetabular walls, ischium and pubic bone as well as the extent of any defects present is one of the most important aspects of planning revision hip arthroplasty. This can be achieved with detailed virtual models of the pelvis, together with 3D printed physical models. However, they should always be treated as additional tools in preparation for hip arthroplasty procedures, and surgeons must still conduct a thorough assessment of their patients, including preoperative radiographs and CT imaging. Ultimately, surgeons will make decisions based on their clinical and surgical experience.

Line 78, the Figure 1 encouraged to giving significant improvement since it is not looks so scientific.

We understand the reviewer's comment, however the image shown is a radiogram of the pelvis with superimposed diagrams that depict the Ranawat method of determining the Center of Rotation (COR). This is the common method of described such methods and routinely used in the literature. Therefore, it would be confusing to most readers, in particular radiologists and orthopedic surgeons to depict this method in any other way.

Please find below an example study in which this type of image has been used:

Olmendo-Garcia N.; Sevilla A. Comparative study of accuracy of Ranawat’s and Pierchon’s methods to determine hip centre with informatics tools. HIP International. 2010;20 (7_suppl):48-51. https://doi.org/10.1177/11207000100200s709

Line 54, can the authors justify the novel and significance in the present technical notes?

In our study we have used methods that have been described before in different studies. However, most of these papers described the use of expensive software and high-end 3D printers. We, on the other hand, have focused on the use of open-source, readily available software with a large online community and numerous online tutorials. This software can be downloaded for free, it is fairly easy to learn and can be used by almost anyone. When used in conjunction with budget desktop 3D printers and relatively cheap anatomical models, it results in a low-cost solution that can be used to perform pre-surgical planning of complex orthopedic procedures. This, in our opinion, is the originality of our research and a novel approach to a well-known problem. Moreover, the workflow we have developed provides a step-by-step guide to preoperative planning using CT data and 3D printed anatomical models.

In line 58-59m since the authors have been used reff. [3] previously, please give additional or replace with another reference as follows: https://doi.org/10.3390/biomedicines11030951, https://doi.org/10.1038/s41598-023-30725-6, and https://doi.org/10.3390/su142013413

According to the Reviewer’s suggestions the appropriate changes have been added to the References.

Please find below the modified fragments of the manuscript:

……by reduced contact pressure and wear of implant surfaces [4-6 9].

Line 104, the patient involved is literally small. It is possibility to added?

The current Technical Note is the first work in a series of studies related to pre-surgical planning of RHA using open-source software and anatomical models built on a low-cost 3D printer. This study focuses on patients with unilateral Paprosky type II defects, due to the fact that such cases are less complex in nature. Thus, we have been able to develop our workflow and methodology for creating 3D models, as well as establishing COR together with inclination and anteversion angles. On the basis of this experience we are currently working on applying these techniques to more complex cases such as unilateral and bilateral Paprosky type III defects.

Line 115, this section is important in the technical note, please more explanation in this section. Line 120, nothing any parameter description? I believe there is other that not author mentioned.

Step 1, three planes were created, the first (Sagittal plane) passing through the center of the pubic symphysis and the center of the sacrum (Figure 2 3 (C)). Next, a second plane was created between the tips of the ischial tuberosities (Axial plane), and finally a third plane (Coronal plane) was formed between the tips of the ischial spines on both sides.

Step 2, a sphere was drawn with dimensions corresponding to the uninjured intact side femur head and its center aligned with the center of the femur head. The distance, at right angles, of the center of this sphere relative to all three planes created in Step 1, was then measured.

Step 3, a second sphere with identical dimensions was drawn on the contralateral side, i.e., the injured revision side, revision acetabulum. This sphere was positioned with its center at identical distances to the three planes, created in Step 1, to those of the sphere on the uninjured intact side side that was aligned with femur head. Thus, a new center of rotation was established, with XYZ coordinates based on the COR of the contralateral uninjured intact side femur head (Figure 2 3 (C)).

Step 4, the inclination and anteversion angles were determined. The next step was to determine the inclination and anteversion angles. The Axial plane was used to determine the inclination angle with values ranging from 30 up to 45 deg. (Figure 2 3 (D)). The Coronal plane was used to determine the angle of anteversion with values ranging from 10 up to 20 deg. (Figure 2 3 (D)).

We have only made minor changes to this part of manuscript in order to clarify the CT study protocol and the number of steps for determining the COR, inclination angle etc. Following the reviewers comments we discussed this section with other medical and engineering personnel. The general consensus is that the key information and parameters, provided in this part of the manuscript, describe the process in sufficient detail. In our opinion it would be difficult to include any other parameters, which are relevant and would provide any significant details. Hence, we have only introduced the minimal changes described above.

Table 1. CT scan parameters [17 23].

Computed Tomography Scanning Protocol

Type of study

Single Phase,
Non-contrast Enhanced Computed Tomography

Region

Pelvis

Patient position

Supine, lower limbs positioned symetrically

Image reconstruction

Reconstruction algorithms (Kernels):

GE: Standard, Bone, Bone Plus;

Siemens: B30f, B30s, B60s, B70s;

Slice Thickness

≤ 1.0 mm (isometric voxel)

Resolution

512x512

Pitch

≤ 1.0

Gantry Tilt Angle

Data Format

Uncompressed DICOM files

 Line 127, for explaining the workflow, please try to give start-finish type with coloured step.

According to the Reviewer’s suggestions the appropriate changes have been made to the workflow

Please find below the modified fragments of the manuscript:

Figure 3. Diagram showing the workflow used in the study. The Post-Op Assessment stage was similar to the Pre-Op stage and consisted of CT data analysis, Segmentation and Virtual Planning.

Round 2

Reviewer 1 Report

I appreciate that all suggested changes have been resolved and the work can be published in its current form.

Reviewer 2 Report

Confirmed author's correction. I believe it has been properly corrected.

I consider the quality of the English language to be adequate.

Reviewer 3 Report

Dear authors, thanks for considering my comments and correcting the manuscript accordingly. Wish you all the best 

Could be improved

Reviewer 4 Report

-

-